# Development of a Scale for Assessing the Meaning of Participation in Care Prevention Group Activities Provided by Local Governments in Japan

**DOI:** 10.3390/ijerph17124499

**Published:** 2020-06-23

**Authors:** Ryu Sasaki, Michiyo Hirano

**Affiliations:** 1Graduate School of Health Sciences, Hokkaido University, Sapporo Hokkaido 060-0812, Japan; r.sasaki.study@gmail.com; 2Faculty of Health Sciences, Hokkaido University, Sapporo Hokkaido 060-0812, Japan

**Keywords:** community, health promotion, preventive care, public health, social activity

## Abstract

The meaning of participation in care prevention group activities may encourage continuous participation, making older adults active and healthy throughout their lives. This study developed a scale to assess the meaning of participation in care prevention group activities. It involved 427 participants in care prevention group activities (CPGAs) in Japan who filled out a self-administered questionnaire between October 2017 and February 2018. The meaning of participation was assessed using 15 items. In total, there were 379 valid responses. A factor analysis yielded two factors: “promotion of self-growth” and “enrichment of daily life”. The goodness of fit index (GFI), comparative fit index (CFI), and root mean square error of approximation (RMSEA) were satisfactory (GFI = 0.923; CFI = 0.960; RMSEA = 0.073). Cronbach’s *α* was 0.939 for the entire scale. The scale scores were significantly correlated with scores of the social activity-related daily life satisfaction scale and Ikigai-9. The scale’s reliability and validity were confirmed, indicating its usability for promoting care prevention efforts.

## 1. Introduction

In order to address global population aging, each local government around the world must promote care prevention among those who are community living. Care prevention is defined as “preventing (delaying) the occurrence of the state requiring long-term care [for] as long as possible, and preventing deterioration as much as possible even in a state requiring long-term care, and further aiming to mitigate a state requiring long-term care” [1]. Specifically, it falls under “health promotion”, “health self-management” or “positive lifestyle behaviors”. In 2015, the aging rates (over 65 years old) were 17.6% and 6.4% in developed and developing countries, respectively; these are expected to increase [2]. There is an urgent need to straggle an aging society with an aging rate (over 65 years old) of 7% or more worldwide. As the ability to conduct activities of daily living (ADL) declines with age [3], care prevention is a universal issue for everyone. It is necessary to create a community which helps to maintain the ADL and encourage people to continue to carry out social activities, even if the ADL decreases, and in which older adults can maintain independence and dignity and live a high quality of life.

However, some older adults do not recognize the need for care prevention, lack knowledge about how to perform care prevention activities, and have difficulty continuing their care prevention activities alone, but do not have a voluntary group nearby. Local government should support older adults to practice care prevention by providing them with the required information, networks, venues, and funds for care prevention.

In Japan, local governments promote care prevention projects, which may provide a model for countries that are aging but do not have clear services for supporting independence. Care prevention group activities (CPGAs) are carried out as part of these projects and include exercise, tea parties, recreation, and hobby activities in general, as well as specialist support activities for rehabilitation [4]. Such activities have been shown to maintain motor function, ADL [5], cognitive function, high compliance in training [6], and quality of life [7] among community-living older adults. Although specialist staff guarantee the effects of activities, local governments aim for activities to be mainly led by participants [4]. Participant-centered activities can develop into continuous participation, which leads to interaction [8], mutual support, and role acquisition for more participants, in addition to voluntary activities not limited to care prevention. If CPGAs are participant-centered, they contribute to the overall health of older adults and are sustainable in the community.

For the promotion of participant-centered CPGAs, the meaning of CPGA participation is important. It is conceptualized by 15 items and involves two factors, namely the “promotion of self-growth” and the “stimulation of psychosomatic and social aspects” [9]. It is associated with continuous participation and a better attitude toward life [9]. Attitude toward life comprises two factors, namely the “enrichment of mental and social aspect” and “health-consciousness”, and denotes the perception of one’s current life [9]. By recognizing the meaning of CPGA participation, that is, its importance and value for oneself, participants will be proactive in CPGAs and not only gain effects directly but also make their entire life active and healthy.

To enable the development of strategies which enhance the meaning of CPGA participation, the concept of meaning must first be quantified. However, no scale has been developed to quantify it. In previous research [9] based on literature reviews of care prevention activities, social activities, and physical activities, items covering participants’ perception of activities that contribute to primary prevention were created. In addition, a hypothesis of the factor structure of the meaning of CPGA participation was established by exploratory factor analysis (EFA) [9]. However, neither confirmatory factor analysis (CFA) nor analyses that examine the relationship with other scales have been conducted, and no validated scale has been created. In this study, we developed a scale for the meaning of participation in CPGAs (the MPCPGA scale). The scale is useful for the evaluation of current activities and the analysis of characteristics, such as activity contents and management methods, that meet the needs. The scale can provide a new viewpoint, one that differs from conventional evaluation based on physical effects of CPGAs.

## 2. Materials and Methods

### 2.1. Operational Definitions

In this study, CPGAs are defined as regular activities targeting community-based older adults and conducted by the specialized staff of local governments. The MPCPGA is defined as the importance and value of CPGAs for older adults, realized through participation in activities, with reference to the definition provided by Miyata et al. [9].

### 2.2. Research Design, Setting, and Participants

This quantitative, descriptive study was conducted in area B of city A, which is a regional center. In 2015, the aging rate (over 65 years old) of city A was 24.9%, and its ratio of older adults in single-person households to all households was 10.4%. City A has the fourth largest population in Japan, the tertiary industry is the main industry and is active, and the enrollment rate after graduating from high school is slightly above the national average. In 2015, the population of area B was approximately 290,000 and its aging rate (over 65 years old) was 24.2%. Area B has various regional characteristics from urban areas to residential areas, and the proportion of older adults with no need for care or support varies from less than 30% to approximately 50% [10].

CPGAs are carried out by combining exercise designed to prevent falls and dementia, recreation, and tea parties for interaction among participants. There are 1067 registrants in area B CPGAs. We did not inform them in advance about this survey and asked all 427 people who voluntarily participated in the activity to cooperate with the survey. Between October 2017 and February 2018, we visited each CPGA venue and administered collective surveys to participants using signed self-administered questionnaires.

### 2.3. Measures

This scale presented in this paper consists of the items created by Miyata et al. [9]. The contents of the items are similar to the recognition of participants revealed by studies interviewing participants in care prevention group activities. In addition, the items have been checked for content validity by a specialized staff member who understands the usual activities and participants of the CPGAs.

For the development and validation of the MPCPGA scale, we selected two external criteria scales and individual factors.

The criterion-related validity was examined by assessing correlations between the two scales and the MPCPGA scale. The scale considered to be more closely related to the MPCPGA scale measures is the social activity-related daily life satisfaction scale among older adult individuals; its reliability and validity were confirmed [11]. For this scale, social activities are defined as all of the voluntary activities conducted during one’s free time, such as interpersonal activities [11]. CPGAs can be classified as a social activity because older adults participate in them voluntarily in the community context. Similar to daily life satisfaction, the concept of meaning of participation includes the importance of CPGAs in the context of participants’ whole lives. Therefore, we considered that a significant positive correlation with this scale would indicate the criterion-related validity of the MPCPGA scale. We also selected the Ikigai-9 scale, which measures the concept of ikigai; its reliability and validity were confirmed [12]. Ikigai is defined as positive feelings for one’s current life, attitudes toward the future, and perceptions of oneself in relation to society [12]. Ikigai has been shown to be related to social activities [13]. Therefore, we hypothesized that the MPCPGA is related to ikigai. A significant positive correlation with this scale was also considered to indicate the criterion-related validity of the MPCPGA scale.

Based on research findings, we selected gender, age, years of residence, family type, certification of the need for long-term care or support, and self-rated health as individual factors. The questionnaire contained 45 items that solicited information on respondents’ individual factors and the years of CPGA participation, the meaning of CPGA participation, social activity-related daily life satisfaction, and ikigai. Family type was classified as living alone, living with a spouse only, or living with spouse/children/grandchildren/others. Certification of the need for long-term care or support was classified as none, support level 1 or 2, or care level 1–5. The need for long-term support and care was characterized by the requirement for continuous care and assistance with basic ADL, such as eating and toileting, due to physical and/or mental disorders. Compared with “support” requirements, “care” requirements reflect the need for more care; higher levels reflect longer-term care needs. Self-rated health was classified as very good, somewhat good, not very good, or poor. The 15 items that assessed the meaning of participation in CPGAs were based on Miyata et al.’s research [9]. Responses to the items were structured by a 5-point scale, from 1 (absolutely not important) to 5 (very important). In previous research, all items had a ceiling effect. Therefore, to reduce bias, we increased the choice of responses and changed the expression of some items in this study. Social activity-related daily life satisfaction was measured using Okamoto’s 14-item scale [11]. Responses to the items were structured by a 5-point scale, from 1 (not at all) to 5 (very much). Higher scores indicate a greater daily life satisfaction obtained from social activities. Ikigai was measured using Imai et al.’s 9-item scale [12]. Responses to the items were structured by a 5-point scale, from 1 (almost not) to 5 (very much). Higher scores indicate a greater sense of ikigai or a more positive attitude toward one’s life.

### 2.4. Analytical Methods

A Kolmogorov–Smirnov test was performed for all measures prior to performing each analysis. As a result, it was confirmed that none of them, except age, followed a normal distribution, so an nonparametric analysis was performed.

First, the response distribution (%), ceiling and floor effects, and inter-item correlation of the items were assessed to examine the constituent items of the MPCPGA scale. Next, Good-Poor (G-P) and Item-Total (I-T) correlation and exploratory factor analyses were performed. Inter-item correlation was assessed for all pairs of items using Spearman rank correlation coefficients. In the G-P analysis, we divided the respondents into low- and high-scoring groups based on the average of the total scores of all items, and then examined differences in average scores for each item between groups using the *t* test. In the I-T correlation analysis, correlations between each item score and the total score for other items were examined using Spearman rank correlation coefficients. In the exploratory factor analyses (using the maximum likelihood method with promax rotation), factors with eigenvalues ≥ 1 were extracted. Items with factor loadings ≥ 0.4 were selected as constituent items of each factor.

To examine the reliability of the scale, Cronbach’s *α* values were calculated for the entire scale and each subscale. Cronbach’s *α* ≥ 0.7 was considered to indicate reliability [14]. To examine the factorial validity, confirmatory factor analysis was performed based on the results of the exploratory factor analysis. This analysis employed a goodness of fit index (GFI), comparative fit index (CFI), and root mean square error of approximation (RMSEA). Goodness of fit was accepted if GFI ≥ 0.9, CFI ≥ 0.95, and RMSEA ≤ 0.08 [15]. To examine the criterion-related validity, correlations between MPCPGA scores and social activity-related daily life satisfaction scale and Ikigai-9 scores were examined using Spearman rank correlation coefficients.

To clarify the relationships between individual factors and the meaning of CPGA participation, we examined differences in MPCPGA scores among subgroups defined by individual factors using the Mann–Whitney *U* or the Kruskal-Wallis test.

IBM SPSS Amos version 23 and IBM SPSS Statistics version 22 (IBM Corp., Armonk, NY, USA) were used for analyses. The significance level for all analyses was set to 5%.

### 2.5. Ethical Considerations

The study was conducted in accordance with the Declaration of Helsinki. We explained the following to the staff in writing, and to the participants in writing and verbally: the study objective, that survey participation was voluntary, assurances of participants’ anonymity, and the collected questionnaire data being password-protected and strictly managed. Participants provided written informed consent. They were informed that consent could be withdrawn at any time. The Ethics Review Committee of the Faculty of Health Sciences, Hokkaido University approved this study (no. 17–80, 11 October 2017).

## 3. Results

Of the 420 returned questionnaires (98.4% response rate), 379 complete and eligible questionnaires (88.8% valid response rate) were analyzed.

### 3.1. Individual Factors

The participant characteristics are shown in Table 1. The study sample comprised 345 (91.0%) women, with a mean (± standard deviation) age of 77.8 ± 6.0 years. The number of participants who lived alone was 136 (36.2%). Most participants had no long-term insurance certification (*n* = 331, 88.7%) and rated their health as somewhat good (*n* = 274, 72.5%). The mean duration of CPGA participation was 4.69 ± 4.5 years.

### 3.2. MPCPGA Response Distribution and Item Parameters

The response distributions and results of MPCPGA item analyses are shown in Table 2. Mean item scores ranged from 3.88 ± 0.99 to 4.35 ± 0.84. The ceiling effect was observed for 13 of the 15 items. The response distribution was considered to be satisfactory, as no item showed a concentration of >60% of responses on one response option. In the G-P analysis, mean scores of the high-scoring group were significantly higher for all items (*p* < 0.001). In the I–T correlation analysis, correlation coefficients were ≥0.5 for all items. Significant inter-item correlations were found between four pairs (*r* > 0.700, *p* < 0.001). The pairs are item 1 and item 2 (r = 0.767), item 1 and item 13 (r = 0.718), item 10 and item 13 (r = 0.741), and item 12 and item 13 (r = 0.728) in Table 2. Finally, we performed the exploratory factor analysis without item 1, “I can change of pace and feel brighter every time”, and item 13, “With this activity, I can add stimulation to my life that I do not experience in daily life”, because these two items showed a particularly high ceiling effect and a strong correlation with other items.

### 3.3. MPCPGA Factors

The results of the exploratory factor analysis are shown in Table 3. Factors 1 (promotion of self-growth) and 2 (enrichment of daily life) comprised nine and four items, respectively. The confirmatory factor analysis (Figure 1) was performed based on the findings of the exploratory factor analysis. Standardized estimation values at the 5% significance level were obtained for all 13 items (range, 0.66–0.82). The goodness of fit values were GFI = 0.923, CFI = 0.960, and RMSEA = 0.073.

### 3.4. Reliability and Validity of the MPCPGA Scale

The Cronbach’s *α* value for the whole MPCPGA scale was 0.939; for factor 1, it was 0.933; and for factor 2, it was 0.838 (Table 3). Significant correlations were observed between scores of the social activity-related daily life satisfaction score and MPCPGA scale (*r* = 0.312, *p* < 0.001), factor 1 (*r* = 0.322, *p* < 0.001) and factor 2 (*r* = 0.223, *p* < 0.001), and between scores of the Ikigai-9 and MPCPGA scale (*r* = 0.457, *p* < 0.001), factor 1 (*r* = 0.488, *p* < 0.001), and factor 2 (*r* = 0.307, *p* < 0.001).

### 3.5. MPCPGA Score Distribution

MPCPGA scores ranged from 13 to 65 (mean = 53.5 ± 9.4; median = 55); factor 1, from 9 to 45 (mean = 37.2 ± 6.8; median = 38); and factor 2, from 4 to 20 (mean = 16.4 ± 3.3; median = 17). Significant gender differences were found in the MPCPGA score (male: median = 52; female: median = 56, *p* < 0.046) and factor 2 score (male: median = 16; female: median = 17, *p* < 0.043). No other individual factor significantly affected the scores.

## 4. Discussion

### 4.1. Performance of the MPCPGA Scale

The MPCPGA scale showed internal consistency, as all Cronbach’s *α* values were ≥0.8. It contained similar items regarding the meaning of CPGA participation and was considered to be reliable. The factorial validity of the scale was also confirmed, as indices showed an adequate fit.

The moderate degree of correlation between MPCPGA and the social activity-related daily life satisfaction scale confirmed that these scales measured similar concepts, but the CPGA was a specialized form of general social activities. The meaning of CPGA participation includes many items that not only contribute to others and society but also enrich one’s life. The moderate degree of correlation between the MPCPGA scale and Ikigai-9 confirmed the relationship between the meaning of CPGA participation and ikigai. Together, these findings indicate that the MPCPGA scale has criterion-related validity.

In this study, 13 of the 15 MPCPGA items showed a ceiling effect. The bias may have arisen from collective or signed surveys aimed at correct understanding of the survey, and traceability. However, scores for the meaning of CPGA participation were similarly high in previous research [9]. CPGAs were considered to be meaningful to participants who continuously participated in the activities. The score distribution would be less biased for a group containing many respondents who had only recently begun to participate.

Although more than 90% of the participants in this study were women, the scale is suitable for target groups engaging in CPGAs or similar activities, such as exercise, as many such groups are predominantly composed of women [5,9,16].

### 4.2. Characteristics of the MPCPGA Scale and Related Factors

Factor 1 of the MPCPGA scale, which is named “promotion of self-growth”, is considered to indicate the meaning associated with the process in which participants gain self-growth by being motivated and actively participating. Those who planned their participation in an exercise group as part of their daily life continuously participated [17]. It is thought to be the basis for older adults to regard CPGA as a regular event in their daily life and continue activities. It helps them become positive, be aware of their challenges, and gain self-growth. It is recognized by participants that CPGAs can provide interaction and expand the experience and knowledge by new learning [18]. Learning activities [19] and interaction with others [20] are related to cognitive function. Participants seem to appreciate that health lectures and consultations with staff and other participants in CPGAs not only spread knowledge, but also stimulate by themselves, helping to prevent dementia. Moreover, taking a role and contributing to other people are related to a higher consciousness of ikigai [13]. Taking a role and utilizing knowledge and experience to contribute to others in CPGAs are thought to lead to the recognition of one’s own existence value and the realization of purpose in life.

Factor 2 of the MPCPGA scale, which is named “enrichment of daily life”, is considered to indicate the meaning regarding the expected impact on daily life that results by simply participating in the program. Social activities were found to enhance one’s relationship with society [21]. Participants involved in regular exercise indicated that it was enjoyable and helped them to maintain their everyday function and increase their independence [22]. Participation in CPGA itself is considered to be an opportunity to relate to society. Additionally, the participants can maintain physical functions, exchange information with other people, and have a good time by participating in activities according to the CPGA program.

Gender was the only individual factor that had a significant effect on MPCPGA scores. The social activities of older adults differ between genders: women tend to have more friendships and frequently communicate with family and friends [23], whereas men tend to actively participate in work-related activities and leisure [24] in which interaction is not the main purpose. Men are less likely to strongly recognize the “enrichment of daily life” through CPGA participation, as this concept stems from interaction with other participants.

The accessibility of CPGA participation explains the lack of an effect of individual factors other than gender on MPCPGA scores. CPGAs are implemented by rehabilitation specialists. Even those who are anxious about their health condition, which is common among later-stage older adults [25], and are experiencing declines in their ADL ability or physical function that tend to limit social activity [26] can participate actively at appropriate and safe levels. In CPGAs, staff members encourage interaction among participants and provide opportunities for exchange. Therefore, even individuals who live alone and have resided in the community for a short time can interact with local people. By realizing the meaning of exercise and interaction in CPGAs, participants can become health conscious and actively exchange with other people, including CPGA participants, in daily life.

### 4.3. Applicability of the MPCPGA Scale

The MPCPGA scale can be utilized widely in the assessment and development of CPGAs managed by specialist staff because it is based on surveys conducted on participants in diverse subdistricts. Based on assessments using this scale, programs that can improve older adults’ participation in care prevention activities and the meaning of participation can be developed and applied, which will lead to continued health promotion. Furthermore, because recognizing the meaning of participation involves understanding its value, it is expected that a new group will be established by older adults themselves (that is, creation of local resources). In addition, by utilizing this scale for various groups, it will be possible to understand general care prevention needs of older adults. Furthermore, as the target group is limited to men and those with a short stay, it is possible to understand the unique needs related to care prevention activities for those groups. Thus, this scale is a useful tool to gain a better understanding of activities that are more meaningful to participants, based on which effective care prevention programs can be developed for diverse groups.

### 4.4. Study Limitations

The predictive validity and stability of the scale could not be examined due to the cross-sectional study. The reliability and validity of the scale could be confirmed more accurately by surveys that include dropouts. In addition, the gender ratio of the study participants was uneven; a more balanced ratio may have yielded different results. To obtain a sufficient number of male participants, further research involving many CPGA venues is needed. Socio-economic factors are not included in the items in this study. Verification of the structure, reliability, and validity of the scale for groups with socio-economic characteristics that differ from the subjects of this study, and an examination of the relationship between socio-economic factors and the meaning of participation, will be conducted in the future. Compared to agricultural and fishing villages, it may be easier to understand the significance of various forms of participation, such as roles in classrooms, consultation with other older adults or staff, and information exchanges in the area where older adults reside, given the connection between residents and the lack of roles in the territorial community.

## 5. Conclusions

This study confirmed the reliability and validity of the MPCPGA scale, which is composed of 13 items under two factors. This scale enables the measurement of the meaning of participation realized by CPGA participants, as well as a consideration of individual factors and activity content. The MPCPGA was found to have a significant relationship with the social activity-related daily life satisfaction and the sense of ikigai. Promoting care prevention from the perspective of the meaning of participation will encourage older adults to have a positive attitude toward life. Not only will it result in continuous care prevention at the individual or community levels, but it will also lead to older adults living active and healthy lives.

## Figures and Tables

**Figure 1 ijerph-17-04499-f001:**
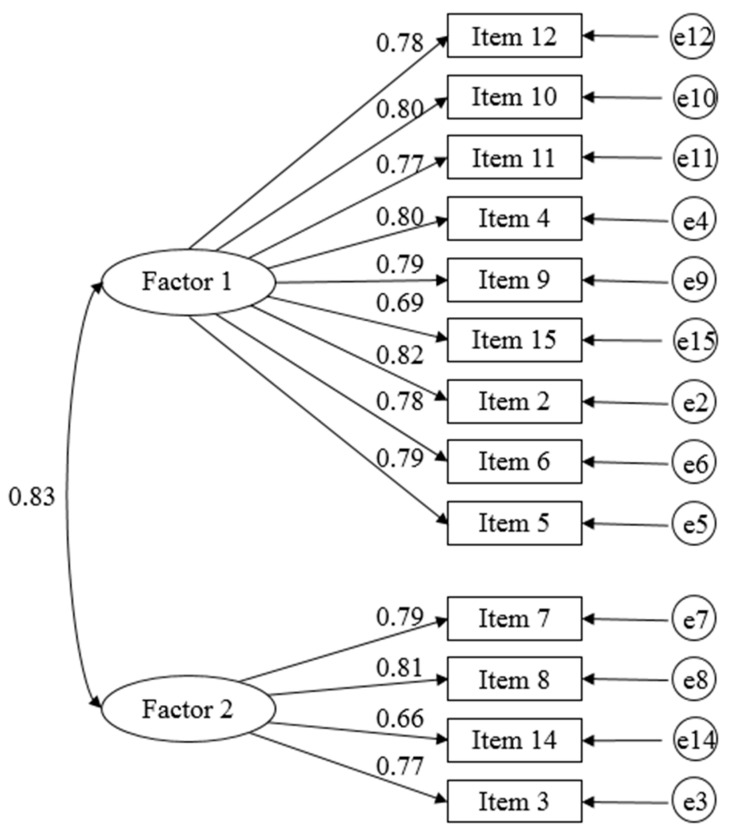
Confirmatory factor analysis: the structure of meaning of participation in care prevention group activities (*n* = 379).

**Table 1 ijerph-17-04499-t001:** Individual factors of subjects (*n* = 379).

Item		Mean ± SD	*n*	(%)
Gender	Male		34	9.0
	Female		345	91.0
Age	65−74	77.8 ± 6.0	111	29.3
	≥75		268	70.7
Years of Residence	<10	28.6 ± 16.7	61	16.1
	10−19		53	14.0
	20−29		81	21.4
	≥30		184	48.5
Family Type ^†^	Alone		136	36.2
	Only spouses		106	28.2
	With spouse, children or grandchildren		132	35.1
	Other		2	0.5
Certification of Needed Long-Term Care or Support ^‡^	Not certificated		331	88.7
	Support level 1		25	6.7
	Support level 2		13	3.5
	Care level 1		3	0.8
	Care level 2		1	0.3
Self-Rated Health ^§^	Very good		27	7.1
	Somewhat good		274	72.5
	Not very good		69	18.3
	Poor		8	2.1
Years of CPGA Participation ^§^		4.69 ± 4.5		

^†^*n* = 376. ^‡^
*n* = 373. ^§^
*n* = 378. SD, standard deviation; CPGA, care prevention group activities.

**Table 2 ijerph-17-04499-t002:** Response distribution and properties of MPCPGA items (*n* = 379).

	Item	Response Distribution (*n* (%))										Item Properties		
		5		4		3		2		1		Mean ± SD	G-P ^†^	I-T
1	I can change of pace and feel brighter every time.	210	(55.4)	98	(25.9)	65	(17.2)	4	(1.1)	2	(0.5)	4.35 ± 0.84	1.18	0.801
2	I can make a regular event to join this enjoyable activity.	207	(54.6)	83	(21.9)	84	(22.2)	4	(1.1)	1	(0.3)	4.30 ± 0.87	1.28	0.802
3	Participation in this activity can be a reason for me to go out.	202	(53.3)	83	(21.9)	82	(21.6)	7	(1.6)	5	(1.3)	4.24 ± 0.94	1.21	0.698
4	I can make use of this activity positively to prevent dementia.	200	(52.8)	85	(22.4)	80	(21.1)	10	(2.6)	4	(1.1)	4.23 ± 0.94	1.30	0.762
5	I can consult with staff or other participants without hesitation.	193	(50.9)	95	(25.1)	76	(20.1)	11	(2.9)	4	(1.1)	4.22 ± 0.94	1.27	0.770
6	I can gain much valuable knowledge and make discoveries.	186	(49.1)	88	(23.2)	88	(23.2)	13	(3.4)	4	(1.1)	4.16 ± 0.97	1.36	0.754
7	I have a satisfying and good time every time.	185	(48.8)	87	(23.0)	95	(25.1)	6	(1.6)	6	(1.6)	4.16 ± 0.96	1.11	0.659
8	I can make use of this activity to maintain or improve physical function.	171	(45.1)	90	(23.7)	104	(27.4)	9	(2.4)	5	(1.3)	4.09 ± 0.97	1.15	0.698
9	I can be positively challenged in this activity.	166	(43.8)	97	(25.6)	95	(25.1)	18	(4.7)	3	(0.8)	4.07 ± 0.97	1.35	0.760
10	This activity can be one of my important purpose in life.	165	(43.5)	103	(27.2)	96	(25.3)	13	(3.4)	2	(0.5)	4.10 ± 0.93	1.33	0.780
11	I can utilize my prior knowledge and experience in this activity.	162	(42.7)	106	(28.0)	101	(26.6)	9	(2.4)	1	(0.3)	4.11 ± 0.90	1.16	0.741
12	Through this activity, I can reflect on my present self and find out my challenges.	160	(42.2)	112	(29.6)	98	(25.9)	5	(1.3)	4	(1.1)	4.11 ± 0.91	1.25	0.744
13	With this activity, I can add stimulation to my life that I do not experience in daily life.	163	(43.0)	108	(28.5)	97	(25.6)	8	(2.1)	3	(0.8)	4.11 ± 0.91	1.37	0.796
14	I can exchange information about our local area with other participants.	149	(39.9)	89	(23.5)	104	(27.5)	26	(6.9)	11	(2.9)	3.89 ± 1.09	1.14	0.563
15	I can actively take a role as a member of this group.	128	(33.8)	107	(28.2)	123	(32.5)	13	(3.4)	8	(2.1)	3.88 ± 0.99	1.16	0.652

^†^ Differences in means between the high-scoring and low-scoring groups (all *p < 0*.001). MPCPGA, the scale of the meaning of participation in care prevention group activities; SD, standard deviation; G-P, good–poor analysis; I-T, item-total analysis.

**Table 3 ijerph-17-04499-t003:** Factor structure of the meaning of participation in care prevention group activities (*n* = 379).

Factor		Item	Factor Loading		Cronbach’s α Total = 0.939
			Factor 1	Factor 2	
Factor 1 (Promotion of Self-Growth)	12	Through this activity, I can reflect on my present self and find out my challenges.	0.884	−0.116	0.933
	10	This activity can be one of my important purpose in life.	0.769	0.049	
	11	I can utilize my prior knowledge and experience in this activity.	0.763	0.017	
	4	I can make use of this activity positively to prevent dementia.	0.740	0.070	
	9	I can positively challenge in this activity.	0.721	0.087	
	15	I can actively take a role as a member of this group.	0.717	−0.032	
	2	I can make a regular event to join this enjoyable activity.	0.690	0.164	
	6	I can gain much valuable knowledge and make discoveries.	0.689	0.118	
	5	I can consult with staff or other participants without hesitation.	0.629	0.195	
Factor 2 (Enrichment of Daily Life)	7	I have a satisfying and good time every time.	−0.105	0.912	0.838
	8	I can make use of this activity positively to maintain or improve physical function.	0.102	0.719	
	14	I can exchange information about our local area with other participants.	0.023	0.649	
	3	Participation in this activity can be a reason for me to go out.	0.248	0.553	
Correlation		Factor 1	1.000	0.752	

Exploratory factor analysis (maximum-likelihood estimation, promax rotation).

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
