# Peer review of "Development of a Scale for Assessing the Meaning of Participation in Care Prevention Group Activities Provided by Local Governments in Japan"

_ijerph, 2020, doi:10.3390/ijerph17124499_

Round 1

Reviewer 1 Report

Thank you for the opportunity to review your manuscript Development of a Scale for Assessing the Meaning of Participation in Care Prevention Group Activities Provided by Local Government in Japan.

I would firstly like to commend you on tackling such an are of inquiry and the lessons that governments more broadly can potentially learn from your research. However, I felt you actually lost sight of the participant in your research. The development of a scale is great work, however, in reviewing your paper there were a number of occasions I wanted to see a mixed methods approach (both quantitative and qualitative) given you are assessing meaning which is a very individual thing. I think you could have taken this aspect further.

Overall I believe your research has merit and I would encourage you to have the manuscript fully edited  for English publication to rectify grammar, work selection and flow issues within the manuscript.

I also would have liked to see more discussion of the findings and as to the relative impact that the scale will have moving forward. The paper is presented factually.

Reviewer 2 Report

Although I like the topic and this investigation is nicely described, i have a few remarks:

  1. in line 30 and 70 you write 'aging rate'; this term is unclear if you don't define it, for example as,  aging rate (%>age 65) 
  2. table 2 is not easy to read because the lines in my version are split in such  a way that all numbers >100 are on the second line 
  3. I miss the description of the representativeness of sample in relation to the population in Japan especially regarding the education and age and income . This sample for example could be characterized by high educated, high income and give a total different outcome compared with a sample that would have a low social economic status. I think the reader should like to know this.

Reviewer 3 Report

First of all, I would like to highlight that the manuscript presents an interesting and relevant topic. Nevertheless, the manuscript requieres modifications to improve for publication.

My comments and questions are the following:

  1. The introduction is a little messy. The first phrase has an authoritarian sense, but the issue emerges from the lack of flow and examples or answering questions.  In this sense, the question why is an obligation for the governments to carry out prevention programs is presented but not really resolve. Another question that pops up is, why is determining the ageing of the population or why is so important these care prevention groups? The authors should highlight the importance of ageing in populations and the relevance of these care prevention groups. Moreover, in some cases the authors have described the sentences without complete depth of the statements. An example would be when the authors indicated  "other countries", in which the idea should be further explained via an example. 
  2. The methodology although is well-presented, there area of improvement. The authors could include a description of the conditions of the area B or the city A, since right now the economic, social or cultural background is missing. Of course the description must be precise and short to avoid the identification of the sample. There is a number 9 as supercript at the line 72 (what this a spelling error?). The authors indicated that 427/1067 were included on the sample, were the 1067 contacted to participate or was used simple random sampling method?. The statistical analysis have focused on non-parametric analysis, was the normalisation studied using the Kolmogorov-Smirnov and being this significant? 
  3. The results are little explained, specially the table 2 and that both contain important information should be further explained. Other thing is the Figure 1, which seems to be screenshot should be improved and the header of the figure should be separated from the screenshot. 
  4. The discussion section is well-structured, although in the limitations the specific factors of the sample should consider, not only the sex but also the social, economic or cultural situation 
  5. The conclusions seems to be a little short since there are relevance data, that should be further highlighted.

Round 2

Reviewer 1 Report

Thank you for the opportunity to re-review your paper.

Firstly congratulations on the significant rework and improvements you have made to both my previous formal comments and also my direct edit comments on version 1 of your paper. I also note that Table 2 is revised and easier to read.

There are only a couple of very minor issues to fix up in version 2.

  1. the title for the figure on page 10 - it seems to have been lost from this version.
  2. there is Japanese writing and other referencing in brackets after Ryu Sasaki on line 5 - this could just be a macro issue
  3. again Japanese writing that needs either translation or removed at line 33.
  4. one last final read for minor typographical issue.

Well done

Reviewer 3 Report

Dear authors,

First of all,I would like to congratulate you for the changes and improvements made on your manuscript.

Nevertheless, I have some minor comments:

  1. Line 33 the reference 4 seems to have words in Japenese.This aso happens in lines 153 ad 154.
  2. In the ethodology section, the description of the smple idicates the rate of graduates, but, considering the main population other data coud be included such as rate of aging.
  3. In the result, the headed of the Figure 1 is missing.
